Discovering generative models from event logs: data-driven simulation vs deep learning

http://orcid.org/0000-0002-8510-1972 Camargo Manuel 1 2 manuel.camargo@ut.ee
http://orcid.org/0000-0002-9247-7476 Dumas Marlon 1
http://orcid.org/0000-0002-8296-6620 González-Rojas Oscar 2
1 Institute of Computer Science, University of Tartu , Tartu , Estonia
2 Computer and Systems Engineering Department, Universidad de Los Andes , Bogotá , Colombia
Ghidini Chiara
Electronic publication date: 2021 Jul 12
Publication date: 2021
Volume: 7
Electronic Location ID: e577
Received 2020 Nov 24; Accepted 2021 May 12
Copyright: © 2021 Camargo et al.
Copyright year: 2021
Copyright holder: Camargo et al.
License: This is an open access article distributed under the terms of the Creative Commons Attribution License, which permits unrestricted use, distribution, reproduction and adaptation in any medium and for any purpose provided that it is properly attributed. For attribution, the original author(s), title, publication source (PeerJ Computer Science) and either DOI or URL of the article must be cited.
License URL: https://creativecommons.org/licenses/by/4.0/

Keywords: Process mining, Deep learning, Data-driven simulation

Funding: European Research Council (PIX project) This work was supported by the European Research Council (PIX project). The funders had no role in study design, data collection and analysis, decision to publish, or preparation of the manuscript.

==============================
A generative model is a statistical model capable of generating new data instances from previously observed ones. In the context of business processes, a generative model creates new execution traces from a set of historical traces, also known as an event log. Two types of generative business process models have been developed in previous work: data-driven simulation models and deep learning models. Until now, these two approaches have evolved independently, and their relative performance has not been studied. This paper fills this gap by empirically comparing a data-driven simulation approach with multiple deep learning approaches for building generative business process models. The study sheds light on the relative strengths of these two approaches and raises the prospect of developing hybrid approaches that combine these strengths.

Introduction

Process mining is a family of techniques that allow users to interactively analyze data extracted from enterprise information systems in order to derive insights to improve one or more business processes. Process mining tools extract business process execution data from an enterprise system and consolidate it in the form of an event log.

In this setting, an event log is a collection of execution traces of a business process. Each trace in an event log consists of a sequence of event records. An event record captures an execution of one activity, which takes place as part of one execution of a business process. For example, in an order-to-cash process, each execution of the process (also known as a case) corresponds to the handling of one purchase order. Hence, in an event log of this process, each trace contains records of the activities that were performed in order to handle one specific purchase order (e.g. purchase order PO2039). This trace contains one event record per activity execution. Each event record contains the identifier of the case (PO2039), an activity label (e.g. “Dispatch the Products”), an activity start timestamp (e.g. 2020-11-06T10:12:00), an activity end timestamp (e.g. 2020-11-06T11:54:00), the resource who performed the activity (e.g. the identifier of a clerk at the company’s warehouse), and possibly other attributes, such as ID of the client.

A generative model of a business process is a statistical model constructed from an event log, which is able to generate traces that resemble those observed in the log as well as other traces of the process. Generative process models have several applications in the field of process mining, including anomaly detection (Nolle, Seeliger & Mühlhäuser, 2018), predictive monitoring (Tax et al., 2017), what-if scenario analysis (Camargo, Dumas & González-Rojas, 2020) and conformance checking (Sani et al., 2020). Two families of generative models have been studied in the process mining literature: Data-Driven Simulation (DDS) and Deep Learning (DL) models.

DDS models are discrete-event simulation models constructed from an event log. Several authors have proposed techniques for discovering DDS models, ranging from semi-automated techniques (Martin, Depaire & Caris, 2016) to automated ones (Rozinat et al., 2009; Camargo, Dumas & González-Rojas, 2020). A DDS model is generally constructed by first discovering a process model from an event log and then fitting a number of parameters (e.g. mean inter-arrival rate, branching probabilities, etc.) in a way that maximizes the similarity between the traces that the DDS model generates and those in (a subset of) the event log.

On the other hand, DL generative models are machine learning models consisting of interconnected layers of artificial neurons adjusted based on input-output pairs in order to maximize accuracy. Generative DL models have been widely studied in the context of predictive process monitoring (Tax et al., 2017; Evermann, Rehse & Fettke, 2017; Lin, Wen & Wang, 2019; Taymouri et al., 2020), where they are used to generate the remaining path (suffix) of an incomplete trace by repeatedly predicting the next event. It has been shown that these models can also be used to generate entire traces (Camargo, Dumas & González-Rojas, 2019) (not just suffixes).

To date, the relative accuracy of these two families of generative process models has not been studied, barring a study that compares DL models vs automatically discovered process models that generate events without timestamps (Tax, Teinemaa & Van Zelst, 2020). This paper fills this gap by empirically comparing these approaches using eleven event-logs, which vary in terms of structural and temporal characteristics. Based on the evaluation results, the paper discusses the relative strengths and potential synergies of these approaches.

The paper is organized as follows. “Generative Data-Driven Process Simulation Models” and “Generative Deep Learning Models of Business Processes” review DDS and DL generative modeling approaches, respectively. “Evaluation” presents the empirical evaluation setup while “Findings” presents the findings. “Discussion” discusses the conceptual trade-offs between DDS and DL approaches in terms of expressiveness and interpretability and relates these trade-offs to the empirical findings. Finally, “Conclusion” concludes and outlines future work.

Generative data-driven process simulation models

Business Process Simulation (BPS) is a quantitative process analysis technique in which a discrete-event model of a process is stochastically executed a number of times, and the resulting simulated execution traces are used to compute aggregate performance measures such as the average waiting times of activities or the average cycle time of the process (Dumas et al., 2018).

Typically, a BPS model consists of a process model enhanced with time and resource-related parameters such as the inter-arrival time of cases and its associated Probability Distribution Function (PDF), the PDFs of each activity’s processing times, a branching probability for each conditional branch in the process model, and the resource pool responsible for performing each activity type in the process model (Dumas et al., 2018). Such BPS models are stochastically executed by creating new cases according to the inter-arrival time PDF, and by simulating the execution of each case constrained to the control-flow semantics of the process model and to the following activity execution rules: (i) If an activity in a case is enabled, and there is an available resource in the pool associated to this activity, the activity is started and allocated to one of the available resources in the pool; (ii) When the completion time of an activity is reached, the resource allocated to the activity is made available again. Hence, the waiting time of an activity is entirely determined by the availability of a resource. Resources are assumed to be eager: as soon as a resource is assigned to an activity, the activity is started.

A key ingredient for BPS is the availability of a BPS model that accurately reflects the actual dynamics of the process. Traditionally, BPS models are created manually by domain experts by gathering data from interviews, contextual inquiries, and on-site observation. In this approach, the accuracy of the BPS model is limited by the accuracy of the process model used as a starting point.

Several techniques for discovering BPS models from event logs have been proposed (Martin, Depaire & Caris, 2016; Rozinat et al., 2009). These approaches start by discovering a process model from an event log and then enhance this model with simulation parameters derived from the log (e.g. arrival rate, branching probabilities). Below, we use the term DDS model to refer to a BPS model discovered from an event log.

Existing approaches for discovering a DDS from an event log can be classified in two categories. The first category consists of approaches that provide conceptual guidance to discover BPS models. For example, Martin, Depaire & Caris (2016) discusses how PM techniques can be used to extract, validate, and tune BPS model parameters, without seeking to provide fully automated support. Similarly, Wynn et al. (2008) outlines a series of steps to construct a DDS model using process mining techniques. The second category of approaches seek to automate the extraction of simulation parameters. For example, Rozinat et al. (2009) proposes a pipeline for constructing a DDS using process mining techniques. However, in this approach, the tuning of the simulation model (i.e., fitting the parameters to the data) is left to the user.

In this research, we use Simod (Camargo, Dumas & González-Rojas, 2020) as a representative DDS method because, to the best of our knowledge, it is the only fully automated method for discovering and tuning business process simulation models from event logs. The use of methods with automated tuning steps, such as that of Rozinat et al. (2009), would introduce two sources of bias in the evaluation: (i) a bias stemming from the manual tuning of simulation parameters, which would have to be done separately for each event log using limited domain knowledge; and (ii) a bias stemming from the fact that the DDS model would be manually tuned while the deep learning models are automatically tuned as part of the model training phase. By using Simod, we ensure a fair comparison, insofar as we compare a DDS method with automatic data-driven tuning of model parameters with deep learning methods that, likewise, tune their parameters (weights) to fit the data. Figure 1 depicts the steps of the Simod method, namely Structure discovery and Time-related parameters discovery.

Figure 1 Pipeline of Simod to generate process models.

In the structure discovery stage, Simod extracts a BPMN model from data and guarantees its quality and coherence with the event log. The first step is the Control Flow Discovery, using the SplitMiner algorithm (Augusto et al., 2019b), which is known for being one of the fastest, simple, and accurate discovery algorithms. Next, Simod applies Trace alignment to assess the conformance between the discovered process model and each trace in the input log. The tool provides options for handling non-conformant traces via removal, replacement, or repair to ensure full conformance, which is needed in the following stages. Then Simod discovers the model branching probabilities offering two options: assign equal values to each conditional branch or computing the conditional branches’ traversal frequencies by replaying the event log over the process model. Once all the structural components are extracted, they are assembled into a single data structure that a discrete event simulator can interpret (e.g., Bimp). The simulator is responsible for reproducing the model at discrete moments, generating an event log as a result. Then Simod uses a hyperparameter optimization technique to discover the configuration that maximizes the Control-Flow Log Similarity (CFLS) between the produced log and the ground truth.

In the time-related parameters discovery stage, Simod takes as input the structure of the optimized model, extracts all the simulation parameters related to the times perspective, and assembles them in a single BPS model. The extracted parameters correspond to the probability density function (PDF) of Inter-arrival times, the Resource pools involved in the process, the Activities durations, the instances generation calendars and the resources availability calendars. The PDFs of inter-arrival times and activities durations are discovered by fitting a collection of possible distribution functions to the data series, selecting the one that yields the smallest standard error. The evaluated PDFs correspond to those supported by the BIMP simulator (i.e., normal, lognormal, gamma, exponential, uniform, and triangular distributions). The resource pool is discovered using the algorithm proposed by Song & Van der Aalst (2008); likewise, the resources are assigned to the different activities according to the frequency of execution. Finally, Simod discovers calendar expressions that capture the resources’ time availability restricting the hours they can execute tasks. Similarly, the tool discovers case creation timetables that limit when the process instances can be created. Once all these simulation parameters are compiled, Simod again uses the hyperparameter optimization technique to discover the configuration that minimizes the Earth Mover’s Distance (EMD) distance between the produced log and the ground truth.

The final product of the two optimization cycles is a model that reflects the structure and the simulation parameters that best represent the time dynamics observed in the ground truth log.

Generative deep learning models of business processes

A Deep Learning (DL) model is a network composed of multiple interconnected layers of neurons (perceptrons), which perform non-linear transformations of data (Hao, Zhang & Ma, 2016). These transformations allow training the network to learn the behaviors/patterns observed in the data. Theoretically, the more layers of neurons there are in the system, the more it becomes possible to detect higher-level patterns in the data thanks to the composition of complex functions (LeCun, Bengio & Hinton, 2015). A wide range of neural network architectures have been proposed in the literature, e.g., feed-forward networks, Convolutional Neural Networks (CNN), Variational Autoencoders (VAE), and Recurrent Neural Networks (RNN). The latter type of architecture is specifically designed for handling sequential data.

DL models have been applied in several sub-fields of process mining, particularly in the context of predictive process monitoring. Predictive process monitoring is a class of process mining techniques that are concerned with predicting, at runtime, some property about the future state of a case, e.g. predicting the next event(s) in an ongoing case or the remaining time until completion of the case.

Figure 2 depicts the main phases for the construction and evaluation of DL models for predictive process monitoring. In the first phase (pre-processing) the events in the log are transformed into (numerical) feature vectors and grouped into sequences, each sequence corresponding to the execution of a case in the process (a trace). Next, a model architecture is selected depending on the prediction target. In this respect, different architectures may be used for predicting the type of the next event, its timestamp, or both. Not surprisingly, given that event logs consist of sequences (traces), various studies have advocated the use of RNNs in the context of predictive process monitoring. The model is then built using a certain training method. In this respect, a distinction can be made between classical generative training methods, which train a single neural network to generate sequences of events, and Generative Adversarial Network (GAN) methods, which train two neural networks by making them play against each other: one network trained to generate sequences and a second network to discriminate between sequences that have been observed in the dataset and sequences that are not present in the dataset. GAN methods have been shown in various applications to outperform classical training methods when a sufficiently large dataset is available, at the expense of higher computational cost.

Figure 2 Phases and steps for building DL models.

Evermann, Rehse & Fettke (2017) proposed an RNN-based architecture with a classical training method to train models that generate the most likely remaining sequence of events (suffix) starting from a prefix of an ongoing case. However, this architecture cannot handle numerical features, and hence it cannot generate sequences of timestamped events. The approaches of Lin, Wen & Wang (2019), and other approaches benchmarked in Tax, Teinemaa & Van Zelst (2020), also lack of this ability to predict timestamps and durations.

In this paper, we tackle the problem of generating traces consisting not only of event types (i.e. activity labels) but also timestamps. One of the earliest studies to tackle this problem in the context of predictive process monitoring was that of Tax et al. (2017), who proposed an approach to predict the type of the next event in an ongoing case, as well as its timestamp, using RNNs with a type of architecture known as Long-Short-Term Memory (LSTM). The same study showed that this approach can be effectively used to generate the remaining sequence of timestamped events, starting from a given prefix of a case. However, this approach cannot handle high dimensional inputs due to its reliance on one-hot encoding of categorical features. As a result, its accuracy deteriorates as the number of categorical features increases. This limitation is lifted in the DeepGenerator approach (Camargo, Dumas & González-Rojas, 2019), which extends the approach of Tax et al. (2017) with two mechanisms to handle high-dimensional input, namely n-grams and embeddings, and integrates a mechanism for avoiding temporal instability namely Random Choice next-event selection. A more recent study, Taymouri et al. (2020), proposes to use a GAN method to train an LSTM model capable of predicting the type of the next event and its timestamp. The authors show that this GAN approach outperforms classical training methods (for the task of predicting the next event and timestamp) on certain datasets.

In the empirical evaluation reported in this paper, we retain the LSTM approach of Camargo, Dumas & González-Rojas (2019) and the GAN approach of Taymouri et al. (2020) as representative methods for training generative DL models from event logs. We selected these methods because they have the capability of generating both the type of the next event in a trace and its timestamp. This means that if we iteratively apply these methods starting from an empty sequence, via an approach known as hallucination, we can generate a sequence of events such that each event has one timestamp (the end timestamp). Hence, these methods can be used to produce entire sequences of timestamped events and therefore they can be used to generate event logs that are comparable to those that DDS methods generate, with the difference that the above DL training methods associate only one timestamp to each event whereas DDS methods associate both a start and an end timestamp to each event. Accordingly, for full comparability, we need to adapt the above two DL methods to generate two timestamps per event. In the following sub-sections we describe each of these approach and how we adapted them to fit this requirement.

DeepGenerator approach

The DeepGenerator approach trains a generative model by using attributes extracted from the original event log, specifically activities, roles, relative times (start and end timestamps), and contextual times (day of the week, time during the day). These generative models are able to produce traces consisting of triplets (event type, role, timestamp). A role refers to a group of resources who are able to perform a given activity (e.g. “Clerk” or “Sales Representative”). In this paper, we adapt DeepGenerator to generate sequences of triplets of the form (event-type, start-timestamp, end-timestamp). Each triplet captures the execution of an activity of a given type (event-type) together with the timeframe during which the activity was executed. In this paper, we do not attach roles to events, in order to make the DeepGenerator method fully comparable to Simod as discussed in “Conclusion”.

In the pre-processing phase (cf. Fig. 2), DeepGenerator applies encoding and scaling techniques to transform the event log depending on the data type of each event attribute (categorical vs continuous). Categorical attributes (activities and roles) are encoded using embeddings in order to keep the data dimensionality low, as this property enhances the performance of the neural network. Meantime, start and end timestamps are relativized and scaled over a range of [0, 1]. The relativization is carried out by first calculating two features: the activities duration and the time-between-activities. The duration of an activity (a.k.a. the processing time) is the difference between its complete timestamp and its start timestamp. The time-between-activities (a.k.a. the waiting time) is the difference between the start timestamp of an activity and the end timestamp of the immediately preceding activity in the same trace. All relative times are scaled using normalization or log-normalization depending on the variability of the times in the event log. Once the features are encoded, DeepGenerator executes the sequences creation step to extract n-grams which allow better handling of long sequences. One n-gram is generated for each step of the process execution and this is done for each attribute independently. Hence, DeepGenerator uses four independent inputs: activity prefixes, role prefixes, relativized durations, and relativized time-between-activities.

In the model training phase, one of two possible architectures is selected for training. These architectures, depicted in Fig. 3, vary depending on whether or not they share intermediate layers. The use of shared layers sometimes helps to better differentiate between execution patterns. DeepGenerator uses LSTM layers or GRU layers. Both of these types of layers are suitable for handling sequential data, with GRU layers sometimes outperforming LSTM layers (Mangal, Joshi & Modak, 2019; Chung et al., 2014).

Figure 3 Explored DeepGenerator architectures: (A) this architecture concatenates the inputs related with activities and roles, and shares the first layer, (B) this architecture completely shares the first layer.

Despite the role’s prefixes are encoded and predicted, their accuracy is not evaluated.

Finally, the post-processing phase uses the resulting DL model in order to generate a set of traces (i.e. an event log). DeepGenerator takes each generated trace and uses the classical hallucination method to repeatedly ask the DL model to predict the next event given the events observed so far (or given the empty trace in the case of the first event). This step is repeated until we observe the “end of trace” event. At each step, the DL model predicts multiple possible “next events”, each one with a certain probability. DeepGenerator selects among these possible events randomly but weighted by the associated probabilities. This mechanism turns out to be the most suitable for the task of generating complete event logs by avoiding getting stuck in the higher probabilities (Camargo, Dumas & González-Rojas, 2019).

LSTM-GAN approach

The approach proposed by Taymouri et al. (2020) trains LSTM generative models using the GAN strategy. The strategy proposed by the authors consists of two LSTM models, one generative and one discriminative, that are trained simultaneously through a game of adversaries. In this game, the generative model has to learn how to confuse a discriminative model to avoid distinguishing real examples from fake ones. As the game unfolds, the discriminative model becomes more capable of distinguishing between fake and real examples, thus forcing the generator to improve the generated examples. Figure 4A presents the general architecture of the GAN strategy proposed in (Taymouri et al., 2020). We performed modifications in every phase of this approach to be able to generate full traces and entire event logs so as to make it fully comparable with DDS methods.

Figure 4 LSTM-GAN architechture: (A) Training strategy, (B) inference strategy.

In the preprocessing phase (cf. Fig. 2), the features corresponding to the activity’s category and relative times are encoded and transformed. The model uses one-hot encoding for creating a binary column for each activity and returning a sparse matrix. We adapted the model to enable the prediction of two timestamps instead of one. The original method by Taymouri et al. (2020) only handles one continuous attribute per event (the end timestamp). We added another continuous attribute to capture the time (in seconds) between the the end of the previous event in the sequence and the start of the current one. This additional attribute is herein called the inter-activity times. Next, the inter-activity times are then rounded up to the granularity of days so as to create a so-called design matrix composed of the one-hot encoded activities and the scaled inter-activity times. Then, we create the prefixes and the expected events in order to train the models. Since the original model was intended to train models starting from a k-sized prefix, all the smaller prefixes were discarded and the prediction of the first event of a trace was not considered. We also adapted the model to be trained to predict zero-size prefixes. For this purpose, we extended the number of prefixes considered by including a dummy start event before each trace and by applying right-padding to the prefixes. This modification of the input implied updating the loss functions in order to consider the additional attribute.

In the model training phase, Taymouri et al. (2020) trained specialized models to predict the next event from prefixes of a predefined size. While this approach is suitable for predicting the next event, it is not suitable for predicting entire traces of unknown size. Therefore, we train a single model with a prefix of size five. This strategy is grounded on the results of the evaluation reported by Sindhgatta et al. (2020), from which the authors concluded that increasing the size of the prefix used by the LSTM models (beyond a size of five events) does not substantially improve the model’s predictive accuracy.

Finally, in the post-processing phase we take the complete predicted suffix to feed back the model instead of considering only the first event predicted by the model (see Fig. 4B). We carry out this operation to take advantage of the fact that the original generative model is a sequence-to-sequence model, which receives a sequence of size k and predicts a sequence of size k. The empirical evidence reported by Camargo, Dumas & González-Rojas (2019) shows that concatenating only the last event predicted by the model generates a rapid degradation in the model’s long-term precision, as the model gets trapped in predicting always the most probable events. Accordingly, we use the random selection to select the next type of event.

Evaluation

This section presents an empirical comparison of DDS and DL generative process models. The evaluation aims at addressing the following questions: what is the relative accuracy of these approaches when it comes to generating traces of events without timestamps? and what is their relative accuracy when it comes to generating traces of events with timestamps?

Datasets

We evaluated the selected approaches using eleven event logs that contain both start and end timestamps. In this evaluation we use real logs from public and private sources and synthetic logs generated from simulation models of real processes:The event log of a manufacturing production (MP) process is a public log that contains the steps exported from an Enterprise Resource Planning (ERP) system (Levy, 2014).

The event log of a purchase-to-pay (P2P) process is a public synthetic log generated from a model not available to the authors.1

The event log from an Academic Credentials Recognition (ACR) process of a Colombian University was gathered from its BPM system (Bizagi).

The W subset of the BPIC2012 (https://doi.org/10.4121/uuid:3926db30-f712-4394-aebc-75976070e91f) event log, which is a public log of a loan application process from a Dutch financial institution. The W subset of this log is composed of the events corresponding to activities performed by human resources (i.e. only activities that have a duration).

The W subset of the BPIC2017 (https://doi.org/10.4121/uuid:5f3067dff10b-45da-b98b-86ae4c7a310b) event log, which is an updated version of the BPIC2012 log. We carried out the extraction of the W-subset by following the recommendations reported by the winning teams participating in the BPIC 2017 challenge (https://www.win.tue.nl/bpi/doku.php?id=2017:challenge).

We used three private logs of real-life processes, each corresponding to a scenario of different sizes of data for training. The POC log belongs to an undisclosed banking process, and the CALL log belongs to a helpdesk process. Both of them correspond to large-size training data scenarios. The INS logs belong to an insurance claims process corresponding to a small size training data. For confidentiality reasons, only the detailed results of these three event logs will be provided.

We used three synthetic logs generated from simulation models of real-life processes (https://zenodo.org/record/4264885). The selected models are complex enough to represent scenarios in which occur parallelism, resource contention, or scheduled waiting times. From these models, we generate event logs varying the number of instances representing greater or lesser availability of training data. The CVS retail pharmacy (CVS) event-log is a large-size training data scenario from a simulation model of an exercise described in the book Fundamentals of Business Process Management (Dumas et al., 2018). We generated the CFM and CFS event logs from an anonymized confidential process. They were used to represent scenarios of large and small size training data.

Table 1 characterizes these logs according to the number of traces and events. The BPI17W and BPI12W logs have the largest number of traces and events, while the MP, CFS and P2P have less traces but a higher average number of events per trace.

Table 1 Event logs description.

Size	Type of source	Event log	Num. traces	Num. events	Num. activities	Avg. activities per trace	Avg. duration	Max. duration	
LARGE	REAL	POC	70,512	415,261	8	5.89	15.21 days	269.23 days	
LARGE	REAL	BPI17W	30,276	240,854	8	7.96	12.66 days	286.07 days	
LARGE	REAL	BPI12W	8,616	59,302	6	6.88	8.91 days	85.87 days	
LARGE	REAL	CALL	3,885	7,548	6	1.94	2.39 days	59.1 days	
LARGE	SYNTHETIC	CVS	10,000	103,906	15	10.39	7.58 days	21.0 days	
LARGE	SYNTHETIC	CFM	2,000	44,373	29	26.57	0.76 days	5.83 days	
SMALL	REAL	INS	1,182	23,141	9	19.58	70.93 days	599.9 days	
SMALL	REAL	ACR	954	4,962	16	5.2	14.89 days	135.84 days	
SMALL	REAL	MP	225	4,503	24	20.01	20.63 days	87.5 days	
SMALL	SYNTHETIC	CFS	1,000	21,221	29	26.53	0.83 days	4.09 days	
SMALL	SYNTHETIC	P2P	608	9,119	21	15	21.46 days	108.31 days	

Evaluation measures

We use a generative process model to generate an event log (multiple times) and then we measure the average similarity between the generated logs and a ground-truth event log. To this end, we define four measures of similarity between pairs of logs: Control-Flow Log Similarity (CFLS), Mean Absolute Error (MAE) of cycle times, Earth-Mover’s Distance (EMD) of the histograms of activity processing times, and Event Log Similarity (ELS). It is important to clarify that the generation of time and activity sequences is not a classification task. Therefore, the precision and recall metrics traditionally used for predicting the next event do not apply. Instead, we use symmetric distance metrics (i.e., that penalize the differences between a and b in the same way as from b to a) that measure both precision and recall at the same time as explained in (Sander et al., 2021).

CFLS is defined based on a measure of distance between pairs of traces: one trace coming from the original event log and the other from the generated log. We first convert each trace into a sequence of activities (i.e. we drop the timestamps and other attributes). In this way, a trace becomes a sequence of symbols (i.e. a string). We then measure the difference between two traces using the Damerau-Levenshtein distance, which is the minimum number of edit operations necessary to transform one string (a trace in our context) into another. The supported edit operations are insertion, deletion, substitution, and transposition. Transpositions are allowed without penalty when two activities are concurrent, meaning that they appear in any order, i.e. given two activities, we observe both AB and BA in the log. Next, we normalize the resulting Damerau-Levenshtein distance by dividing the number edit operations by the length of the longest sequence. We then define the control-flow trace similarity as the one minus the normalized Damerau-Levenshtein distance. Given this trace similarity notion, we pair each trace in the generated log with a trace in the original log, in such a way that the sum of the trace similarities between the paired traces is maximal. This pairing is done using the Hungarian algorithm for computing optimal alignments (Kuhn, 1955). Finally, we define the CFLS between the real and the generated log as the average similarity of the optimally paired traces.

The cycle time MAE measures the temporal similarity between two logs. The absolute error of a pair of traces T1 and T2 is the absolute value of the difference between the cycle time of T1 and that of T2. The cycle time MAE is the mean of the absolute errors over a collection of paired traces. Like for the CFLS measure, we use the Hungarian algorithm to pair each trace in the generated log with a corresponding trace in the original log.

The cycle time MAE is a rough measure of the temporal similarity between the traces in the original and the generated log. It does not take into account the timing of the events in a trace—only the cycle time of the full trace. To complement the cycle time MAE, we use the Earth Mover’s Distance (EMD) between the normalized histograms of the mean durations of the activities in the ground-truth log vs the same histogram computed from the generated log. The EMD between two histograms H1 and H2 is the minimum number of units that need to be added to, removed to, or transferred across columns in H1 in order to transform it into H2. The EMD is zero if the observed mean activity durations in the two logs are identical, and it tends to one the more they differ.

The above measures focus either on the control-flow or on the temporal perspective. To complement them, we use a measure that combines both perspectives, namely the ELS as defined in Camargo, Dumas & González-Rojas (2020). This measure is defined in the same way as CLFS above, except that it uses a distance measure between traces that takes into account both the activity labels and the timestamps of activity labels. This distance measure between traces is called Business Process Trace Distance (BPTD). The BPTD measures the distance between traces composed of events that occur in time intervals. This metric is an adaptation of the CFLS metric that, in the case of label matching, assigns a penalty based on the differences in times. BPTD also supports parallelism, which commonly occurs in business processes. To do this, BPTD validates the concurrency relationship between activities applying the oracle used by the alpha algorithm in process discovery. We have called ELS the generalization of the BPTD that measures the distance between two event logs using the Hungarian algorithm (Kuhn, 1955).

Experiment setup

The aim of the evaluation is to compare the accuracy of DDS models vs DL models discovered from event logs. Figure 5 presents the pipeline we followed.

Figure 5 Experimental pipeline.

We used the hold-out method with a temporal split criterion to divide the event logs into two folds: 80% for training and 20% for testing. Next, we use the training fold to train the DDS and the DL models. The use of temporal splits is common in the field of predictive process monitoring (from which the DL techniques included in this study are drawn from) as it prevents information leakage (Camargo, Dumas & González-Rojas, 2019; Taymouri et al., 2020).

We use the first 80% of the training fold to construct candidate DDS models and the remaining 20% for validation. We use Simod’s hyperparameter optimizer to tune the DDS model (see the tool’s two discovery stages in “Generative Data-Driven Process Simulation Models”). First, the optimizer in the structure discovery stage was set to explore 15 parameter configurations with five simulation runs per configuration. At this stage, we kept the DDS model that gave the best results on the validation sub-fold in terms of CFLS averaged across the five runs. Second, the optimizer in the time-related parameters discovery stage was set to explore 20 parameter configurations with five simulation runs per configuration. Then, we hold the DDS model that gave the best results on the validation sub-fold in terms of EMD averaged across the five runs. As a result of the two stages, Simod found the best model in both structure and time dynamics. We defined the number of optimizer trials in each stage, by considering the differences in the search space’s size in each stage (see Simod’s model parameters in Table 2).

Table 2 Parameter ranges and distributions used for hyperparameter optimization.

Model	Stage	Parameter	Distribution	Values	
Simod	Structure discovery	Parallelism threshold (ε)	Uniform	[0…1]	
Percentile for frequency threshold (η)	Uniform	[0…1]	
Conditional branching probabilities	Categorical	{Equiprobable, Discovered}	
Time-related parameters discovery	Log repair technique	Categorical	{Repair, Removal, Replace}	
Resource pools similarity threshold	Uniform	[0…1]	
Resource availability calendar support	Uniform	[0…1]	
Resource availability calendar confidence	Uniform	[0…1]	
Instances creation calendar support	Uniform	[0…1]	
Instances creation calendars confidence	Uniform	[0…1]	
LSTM/GRU	Training	N-gram size	Categorical	{5, 10, 15}	
Input scaling method	Categorical	{Max, Lognormal}	
# units in hidden layer	Categorical	{50, 100}	
Activation function for hidden layers	Categorical	{Selu, Tanh}	
Model type	Categorical	{Shared Categorical, Full Shared}	

The experimental results shows that the best possible value is reached in fewer attempts than expected. Figure 6 shows the log P2P in which the best model was found in the first optimization stage at the trial 10 and in the second stage at the trial 13.

Figure 6 Bayesian hyperparameter optimizer trials: (A) In CFLS units the higher the best, (B) in EMDunits the lower the best.

Next, we apply random search for hyperparameter optimization for each family of generative models (LSTM and GRU). Similarly to the DDS approach, we explore 40 random configurations with five runs each, using 80% of the training fold for model construction and 20% for validation. This sample size was chosen to ensure a confidence level of 95 % with a confidence interval of 6 (see LSTM/GRU’s model parameters in Table 2).

In the case of the LSTM-GAN implementation, as proposed by the authors (Taymouri et al., 2020), we dynamically adjust the size of hidden units in each layer being twice the input’s size. Additionally, we use 25 training epochs, a batch of size five, and a prefix size of five.

The above led us to one DDS, one LSTM, one GRU, and one LSTM-GAN model per log. We then generated five logs per retained model. To ensure comparability, each generated log was of the same size (number of traces) as the testing fold of the original log. We then compare each generated log with the testing fold using the ELS, CFLS, EMD and MAE measures defined above. We report the mean of each of these measures across the 5 logs generated from each model in order to smooth out stochastic variations.

Findings

Figure 7 presents the evaluation results of CFLS, MAE and ELS measures grouped by event log size and source type. Table 3 presents the exact values of all metrics sorted by metric, event log size, and source type. The Event-log column identifies the evaluated log; meanwhile, the GRU, LSTM, LSTM (GAN), and SIMOD columns present the accuracy measures. Note that ELS and CFLS are similarity measures (higher is better), whereas MAE and EMD are error/distance measures (lower is better).

Figure 7 Evaluation results: In the first column the CLFS are presented in similarity units (the higherthe better), the second column presents the MAE results in distance units, and the third column presents the ELS results (the higher the better).

Table 3 Detailed evaluation results.

In bold best accuracy values. CLFS and ELM metrics are similarity measures the biggest the best, MAE and EMD are distance measures the lowest the best.

Metric	Size	Type of source	Event log	GRU	LSTM	LSTM–GAN	SIMOD	
CLFS	LARGE	REAL	POC	0.63141	0.67176	0.28998		
		BPI17W	0.63751	0.71798	0.36629	0.58861	
		BPI12W	0.58375	0.70228	0.35073	0.53744	
		CALL	0.82995	0.83043	0.24055	0.62911	
	SYNTHETIC	CVS	0.83369	0.85752	0.20898	0.71359	
		CFM	0.81956	0.60224	0.11412	0.77094	
SMALL	REAL	INS	0.50365	0.51299	0.25619	0.61034	
		ACR	0.78413	0.78879	0.18073	0.67959	
		MP	0.27094	0.23197	0.06691	0.34596	
	SYNTHETIC	CFS	0.69543	0.66782	0.10157	0.76648	
		P2P	0.41179	0.65904	0.13556	0.45297	
MAE	LARGE	REAL	POC	801147	778608	603105		
			BPI17W	868766	603688	828165	961727	
			BPI12W	701892	327350	653656	662333	
			CALL	160485	174343	159424	679847	
		SYNTHETIC	CVS	859926	667715	952004	1067258	
			CFM	25346	15078	956289	252458	
	SMALL	REAL	INS	1586323	1516368	1302337	1090179	
			ACR	344811	341694	296094	230363	
			MP	335553	321147	210714	298641	
		SYNTHETIC	CFS	30327	33016	717266	15297	
			P2P	2407551	2495593	2347070	1892415	
ELS	LARGE	REAL	POC	0.58215	0.65961	0.28503		
			BPI17W	0.63643	0.70317	0.35282	0.58412	
			BPI12W	0.57862	0.67751	0.33649	0.52555	
			CALL	0.79336	0.81645	0.19123	0.59371	
		SYNTHETIC	CVS	0.65160	0.70355	0.16854	0.70154	
			CFM	0.68292	0.43825	0.09505	0.66301	
	SMALL	REAL	INS	0.49625	0.50939	0.23070	0.57017	
			ACR	0.75635	0.45737	0.15884	0.71977	
			MP	0.25019	0.21508	0.04570	0.31024	
		SYNTHETIC	CFS	0.54433	0.57392	0.07930	0.67526	
			P2P	0.22923	0.39249	0.09968	0.43202	
EMD	LARGE	REAL	POC	0.00036	0.00011	0.00001		
			BPI17W	0.00060	0.01010	0.00072	0.00057	
			BPI12W	0.00077	0.00061	0.00006	0.00002	
			CALL	0.00084	0.15794	0.00090	0.00072	
		SYNTHETIC	CVS	0.61521	0.57217	0.40006	0.13509	
			CFM	0.00472	0.00828	0.03529	0.06848	
	SMALL	REAL	INS	0.03343	0.00308	0.33336	0.00001	
			ACR	0.49996	0.68837	0.25012	0.58674	
			MP	0.12609	0.33375	0.28577	0.31411	
		SYNTHETIC	CFS	0.08253	0.10784	0.06924	0.03461	
			P2P	0.25306	0.33747	0.23898	0.03888	

The results show a clear dependence of training data size on the models’ accuracy. For small logs, Simod presents a greater similarity in the control flow generation in three of the five evaluated logs as shown by the CFLS results. In the remaining two logs, the measure is not far from the best-reported values. In terms of MAE, Simod obtains the smallest errors in four of the five logs, which leads to greater ELS similarity in four of the five logs. However, for large logs, the LSTM model presents the best CFLS results in five of the six evaluated logs wheres the GRU model approaches better in the remaining one. In terms of MAE, the LSTM model obtains the lowest errors in four of the six logs, whereas the LSTM-GAN model approaches better in the remaining two. The difference between the DL and Simod models for the MAE mesure is constant, and dramatic in some cases such as the CALL log. In this log, Simod generates a difference almost forty times greater than that reported by the DL models. This can be the result of a contention of resources that is non-existent in the ground truth.

When analyzing the ELS measure, which joins the two perspectives of control flow and time distance, the LSTM model obtains the greatest similarity in five out of six models and the GRU model in the remaining one. The LSTM-GAN model does not obtain a better result in this metric due to its poor performance in control flow similarity. The LSTM-GAN model’s low performance is because the temporal stability of the models’ predictions declines rapidly, despite having a higher precision in predicting the next event as demonstrated in (Taymouri et al., 2020). This result also indicates overfitting on the models preventing the generalization of this approach for this predictive task.

On the one hand, the results indicate that DDS models perform well when capturing the occurrence and order of activities (control-flow similarity), and that this behavior is independent of the training dataset size. A possible explanation for this result is that event logs of business processes (at least the ones included in this evaluation) follow certain normative pathways captured sufficiently by automatically discovered simulation models. However, Deep Learning models and especially LSTM models outperform the DDS models if a sufficiently large training dataset is available.

On the other hand, Deep Learning models are more accurate when it comes to capturing the cycle times of the cases in the large logs (cf. the lower MAE for DL models vs DDS models). Here, we observe that both DDS and DL models achieve similar EMD values, which entails that both types of models predict the processing times of activities with similar accuracy. Therefore, we conclude that the differences in temporal accuracy (cycle time MAE) between DL and DDS models come from the fact that DL models can better predict the waiting times of activities, rather than the processing times.2

The inability for DDS models to accurately capture the waiting times can be attributed to the fact that these models rely on the assumption that the waiting times can be fully explained by the availability of resources. In other words, DDS models assume that resource contention is the sole cause of waiting times. Furthermore, DDS models operate under the assumption of eager resources as discussed in “Generative Data-Driven Process Simulation Models” (i.e., resources start an activity as soon as it is allocated to them). Conversely, DL models try to find the best possible fit for observed waiting times without any assumptions about the behavior of the resources involved in the process.

Discussion

The results of the empirical evaluation reflect the trade-offs between DDS models and deep learning models. Indeed, these two families of models strike different tradeoffs between modeling capabilities (expressive power) on the one hand, and interpretability on the other.

The results specifically put into evidence the limitations in modeling capabilities of DDS models. Such limitations arise both along the control-flow level perspective (sequences of events) and along the temporal perspective (timestamps associated to each event).

From a control-flow perspective, DDS models can only generate sequences that can be fully parsed by a business process model. In the case of Simod, this model is a BPMN model. The choice of modeling notation naturally introduces a representational bias (Van der Aalst, 2011). For example, free-choice workflow nets—which have the expressive power of BPMN models with XOR and AND gateways (Favre, Fahland & Völzer, 2015)—have limitations that prevent them from capturing certain synchronization constructs (Kiepuszewski, Ter Hofstede & Van der Aalst, 2003). Adopting a more expressive notation may reduce this representational bias, possibly at the expense of interpretability. Furthermore, any DDS approach relies on an underlying automated process discovery algorithm. For example, Simod relies on the Split Miner algorithm (Augusto et al., 2019b) to discover BPMN models. Every such algorithm is limited in terms of the class of process models that it can generate. For example the Split Miner and other algorithms based on directly-follows graphs (e.g. Fodina) cannot capture process models with duplicate activity labels (i.e. multiple activity nodes in the model sharing the same label). Meanwhile, the Inductive Miner algorithm cannot capture non-block-structured process models (Augusto et al., 2019a). In contrast, deep learning models for sequence generation rely on non-linear functions that model the probability that a given activity occurs after a given sequence prefix. Depending on the type of architecture used and the parameters (e.g. the number of layers, the type of activation function, learning rate), these models may be able to learn dependencies that cannot be captured by the class of BPMN models generated by a given process discovery algorithm such as Split Miner.

Along the temporal perspective, DDS models make assumptions about the sources of waiting times of activities. Chiefly, DDS models assume that waiting times are caused exclusively by resource contention and they assume that as soon as a resource is available and assigned to an activity, the resource will start the activity in question (robotic behavior) (Van der Aalst, 2015). Furthermore, DDS models generally fail to capture inter-dependencies between multiple concurrent cases (besides resource contention) such as batching or prioritization between cases (some cases having a higher priority than others) (Van der Aalst, 2015). Another limitation relates to the assumption that resources perform one activity at a time, i.e. no multi-tasking (Estrada-Torres et al., 2020). In contrast, deep learning models simply try to learn the time to the next-activity in a trace based on observed patterns in the data. As such, they may learn to predict delays associated with inter-case dependencies as well as delays caused by exogenous factors such as workers being busy performing work not related to the simulated process. These observations explain why deep learning models outperform DDS models when it comes to capturing the time between consecutive activities (and thus the total case duration). DDS models are prone to underestimating waiting times, and hence cycle times, because they only take into account waiting times due to resource contention. Meanwhile, deep learning models learn to replicate the distributions of waiting times regardless of their origin.

On the other hand, that DDS models are arguably more interpretable than deep learning models, insofar as they rely on a white-box representation of the process that analysts typically use in practice. This property implies that DDS models can be modified by business analysts to capture what-if scenarios, such as what would happen is a task was removed from the model. Also, DDS models explicitly capture one of the possible causes of waiting times, specifically resource contention, while deep learning models do not explicitly capture any such mechanism. As such, DDS models are more amenable to capture increases or reductions in waiting or processing times that arise when a change is applied to a process. Specifically DDS models are capable of capturing the additional waiting time (or the reduction in waiting time) that result from higher or lower resource contention, for example due to an increase in the number of cases created per time unit.

Conclusion

In this paper, we compared the accuracy of two approaches to discover generative models from event logs: Data-Driven Simulation (DDS) and Deep Learning (DL). The results suggest that DDS models are suitable for capturing the sequence of activities of a process. On the other hand, DL models outperform DDS models when predicting the timing of activities, specifically the waiting times between activities. This observation can be explained by the fact that the simulation models used by DDS approaches assume that waiting times are entirely attributable to resource contention, i.e. to the fact that all resources that are able to perform an enabled activity instance are busy performing other activity instances. In other words, these approaches do not take into account the multitude of sources of waiting times that may arise in practice, such as waiting times caused by batching, prioritization of some cases relative to others, resources being involved in other business processes, or fatigue effects.

A natural direction for future work is to extend existing DDS approaches in order to take into account a wider range of mechanisms affecting waiting times, so as to increase their temporal accuracy. However, the causes of waiting times in business processes may ultimately prove to be so diverse, that no DDS approach would be able to capture them in their entirety. An alternative approach would be to combine DDS approaches with DL approaches so as to take advantage of their relative strengths. In such a hybrid approach, the DDS model would capture the control-flow perspective, while the DL model would capture the temporal dynamics, particularly waiting times. The DSS model would also provide an interpretable model that users can change in order to define “what-if” scenarios, e.g. a what-if scenario where an activity is removed or a new activity is added.

Two challenges need to be overcome to design such a hybrid DDS-DL approach: (i) how to integrate the DDS model with the DL model; and (ii) how to incorporate the information of a what-if scenario into the DL model.

A possible approach to tackle the first of these challenges is to generate sequences of events using a DDS model, or more specifically a stochastic model trained to generate distributions of sequences of activities (Sander et al., 2021). In a second stage, the traces generated by such a stochastic model can be extended to incorporate timestamps via a deep learning model, trained to predict waiting times, i.e. the time between the moment an activity is enabled and the time it starts. Processing times can then be added using either a deep learning model, or a temporal probability distribution as in DDS approaches.

To tackle the second of the above challenges, we need a mechanism to adjust the predictions made by a deep learning model in order to capture a change in the process, e.g. the fact that an activity has been deleted. A possible approach is to adapt existing techniques to incorporate domain knowledge (e.g. the fact that an activity will not occur in a given suffix of a trace) into the output of a DL model (Di Francescomarino et al., 2017).

In this paper, we focused on comparing DDS and DL approaches designed to predict sequences of activities together with their start and end timestamps. An event log may contain other attributes, most notably the resource who performs each activity and/or the role of this resource. A possible direction for future work is to compare the relative performance of DDS and DL approaches for the task of generating event logs with resources and/or roles. While there exist deep learning approaches to generate sequences of events with resources (Camargo, Dumas & González-Rojas, 2019), existing DDS approaches, including Simod, are not able to discover automatically tuned simulation models covering the resource perspective. To design such a DDS approach, we need to first define a loss function that takes into account both the control-flow and the resource perspectives. We also need to incorporate a mechanism to assign a specific (individually identified) resource to each activity instance, while ensuring that the associations between activity instances and resources in the simulated log are reflective of those observed in the original log. In other words, a possible direction for future work is to design a DDS technique that handles roles and resources as first-class citizens and to compare the relative performance of such a DDS technique against equivalent DL techniques.

Reproducibiltiy package: Table 4 links to the repositories of the approaches used in the evaluation. The datasets, generative models, and the raw and summarized results can be found at: DOI 10.5281/zenodo.4699983.

Table 4 Source code of the approaches used in the evaluation.

Approach	Repository	
Simod tool	https://github.com/AdaptiveBProcess/Simod.	
DeepGenerator	https://github.com/AdaptiveBProcess/GenerativeLSTM.	
Adapted LSTM(GAN)	https://github.com/AdaptiveBProcess/LSTM-GAN.	

Additional Information and Declarations

Competing Interests

Author Contributions

Data Availability

1 The log is provided as part of the Fluxicon Disco tool—https://fluxicon.com/.

2 The cycle time of a process instance adds the processing times (activity durations) and the waiting times (Dumas et al., 2018).

The authors declare that they have no competing interests.

Manuel Camargo conceived and designed the experiments, performed the experiments, analyzed the data, performed the computation work, prepared figures and/or tables, and approved the final draft.

Marlon Dumas conceived and designed the experiments, authored or reviewed drafts of the paper, and approved the final draft.

Oscar González-Rojas conceived and designed the experiments, authored or reviewed drafts of the paper, and approved the final draft.

The following information was supplied regarding data availability:

The Reproducibility package for the DDS vs DDL article is available at GitHub: https://github.com/AdaptiveBProcess/DDSvsDL.

The datasets, generative models, and the raw and summarized results are available at Zenodo: Manuel Camargo, Marlon Dumas, & Oscar González-Rojas. (2021, April 19). Reproducibility package for “Discovering Generative Models from Event Logs: Data-driven Simulation vs Deep Learning” (Version 0.1.0). Zenodo. DOI 10.5281/zenodo.4699983.

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
