# Peer review of "Discovering generative models from event logs: data-driven simulation vs deep learning"

_PeerJ Computer Science, doi:10.7717/peerj-cs.577_

## Round 0.1 · original submission · Major Revisions

Thank you for this submission. Please find attached the comments, suggestions and questions of the three reviewers. Of particular importance:

- both R2 and R3 have concerns on the limited range of DL methodologies used. Please make an attempt to substantially revise your work adding further DL methodologies taking into account the suggetsins of R3.

- related to the above, please make sure that our conclusions refer to the scope of your evaluation (either by tunjng the conclusions or extending the evaluation).

- R1 is asking for a model to model comparison, especially concerning their expressivity.

- Please consider extending the investigation to recall like metrics as asked by R3.

There are many other points to address raised by the reviewers in their comments that deserve consideration.

I hope these observations will help improve the paper.

Best regards,
The Associate Editor

Reviewer 1 ·

Basic reporting

The article presents an empirical comparison of two approaches for process discovery/predictive monitoring. The first approach is based on simulation mining: a detailed simulation process model (e.g., a Petri net) is discovered from the data. Then, the model is enriched and used for the analysis. The second approach is based on the deep learning - a generative neural network is trained from the data. Both models are essentially trace generators (including rich features that can be generated into the future). The empirical evaluation is very nice and the topic is worth investigating. The paper is very relevant for practitioners and scientists who are working in the area of process mining. One of the novelties of the approach is the realizations that the two representations (simulation and deep learning) have equivalent usages.

What is missing in my opinion is a comparison of their expressive power. Given that the same event log is used to create both - do these two models actually capture the same amount of information? Are they equal in their modeling strength?

I am missing a model-to-model comparison to complement the nice empirical evaluation.

In terms of the standards: the paper is well-written, the literature is covered in a sufficient manner, the paper is structured professionally, is self-contained, and the results are presented in a satisfying manner.

Experimental design

The experimental results are well-aligned with the journal's Aims and Scope. The research questions are well defined, and the paper does fill in a gap in the literature. It is the first paper that considers this equivalence between two generative families of models. I would also like to see a more rigorous comparison of the two models: how are the building blocks compare to each other? are the models equivalent? is one more expressive than the other?

Otherwise, I have no complaints regarding the empirical evaluation.

Validity of the findings

The strengthen the validity of the results, the authors must compare between the two models- this must at least be a discussion, if not a formal comparison between the two languages that the models can generate, and the random variables that underlie the two models.

Conclusions and limitations are well-stated and the empirical evaluation is described in a valid and satisfying manner.

Additional comments

For me to vote accept you must provide a discussion or a formal section (depending on the depth and the comparability of the two approaches) that would satisfy the gap in expressiveness. I have the intuition that the comparison will be interesting and useful for the scientific community. Furthermore, that would really strengthen the novelty and the rigor of the paper.

Reviewer 2 ·

Basic reporting

The language used is clear and mostly unambiguous. However, DDS/SIMOD and DL/LSTM/GRU are used in an interchangeable way. DL and DDS have a higher level of abstraction than SIMOD/LSTM/GRU, whereas in a number of situations - which I will later report in detail in my minor concerns - the authors use them interchangeably.
The work is structured in a reasonable way.
The figures are of high quality and the captions are accurate. I would suggest some minor adjustments to some captions and tables which I will report in my minor concerns.
The raw data are supplied and available.

Experimental design

The scope of the work is within the scope of the journal.
The research questions are well stated and meaningful.
The methods are described with high detail but some meta-parameters for the tuning of hyperparameters could benefit a better description/motivation.

Validity of the findings

The work impact for the field is important and the novelty is well assessed.
The statistical significance of the results could not be assessed due to a motivated shortage in the amount of datasets available in the literature. The results are sound, but the control should be improved due to a not well motivated usage of meta-parameters for the the tuning of hyperparameters.
The conclusions are sound and coherent with the rest of the work.

Additional comments

In the work “Discovering generative models from event logs: data-driven simulation vs deep learning” the authors aim to empirically compare Data-Driven Simulation (DDS) and Deep Learning (DL) approaches for the discovery of generative model from event logs.
The DDS approach of choice is SIMOD and the DL approach of choice is DeepGenerator. In turn Deep Generator uses two different architectures, one is exploiting LSTMs and one GRUs.
The three approaches are compared on 5 different datasets, one of which is synthetic, and four are real-life datasets.
Both DDS and DL approaches are tuned automatically exploiting hyperparameter optimisation techniques.
The discovered generative model is capable of generating a new dataset that resembles the original one.
The quality of the retrieved new dataset and therefore of the generative model is evaluated through 4 different metrics.
The results suggest that DDS approaches are suitable for capturing the sequence of activities/categorical attributes of a process. Whereas DL approaches are suitable for predicting the timing of the activities. These results raise the prospect of combining the two approaches into hybrid techniques.


I think that the paper needs some refinements but it is interesting and timely for the field.
The work is well written and pretty clear, nonetheless it could benefit of a number of adjustments.

The major concerns I have regarding your work are the following ones:
1. The methodologies you evaluate in your work are limited to specific implementations of one DDS and two DL approaches. In the threats of validity you claim that the other state-of-the-art DDS methodologies require manual intervention to be tuned, and that the other available DL methodologies are left as future work. 
While I am fine with leaving some of the DL methods as future work and I believe that SIMOD is an extremely innovative approach, I would ask you that:
 a. either you better motivate why you only use (these) three methodologies (and this should be clear from the beginning), while in your title and introduction you claim to be comparing the DDS and DL families of approaches for the discovery of generative models;
 b. or you assess at least another DDS method in order to provide more general results.
2. The meta parameters used to optimise the hyper parameters of your approach, i.e., the number of configurations and the runs carried out for each configuration, are not well stated/motivated. For the DDS method you state to explore 50 random configurations with 5 runs each.
It seems to be a big amount of models to explore, how did you come up with these two parameters (i.e. 50 and 5)? 
On the other hand, for the DL method you state you use random search to determine the hyperparameters. 
Are there any meta parameters for the random search? If so, can you report and motivate them (e.g. the maximum amount of runs, the maximum time elapsed or the convergence of the retrieved models)?
3. Roles are a very peculiar and interesting sort of data to be used in process mining. While you motivated why you do not use them in Section 3, you do not mention them in Section 2. How would the investigated techniques behave with the generation of event logs not only with timestamps but also with attributes as roles? 
I would ask:
 a. either you clarify why you do not take into account the generation of event logs enriched with roles (e.g., SIMOD does not support the generation of roles)
b. or experiment and compare the techniques of the two families of approaches for the generation of event logs with timestamps and roles.

Other than that, there is a number of minor concerns - mostly regarding the coherence of the presentation of your work - which I would appreciate if you could address:
- Line 96: would it be possible to introduce why you use SIMOD, like discussed in the threats to validity at Line 326
- Figure 1: the caption reports SIMOD instead of DDS, whereas in Figure 2 the label reports DL instead of LSTM and GRU, I would prefer to see the two label aligned
- Figure 4: in the label “SIMOD parameter extraction” could you align to the other label by writing it as “DDS parameter extraction”?
- Table 2 could you rewrite DDS as SIMOD, as you did for DL, reporting them as GRU and LSTM?
- Line 57: would it be possible to report a measure of difference between the datasets? e.g., using ELS between the datasets
- Line 187: would it be possible to provide more details on the reason to use DeepGenerator?
- Line 195: What do you mean by: “DeepGenerator selects among these possible events randomly, taking into account the associated probabilities”? Is it a random or does it use probabilities?
- Line 62: The title “Data-driven simulation” is not in line with the title “Generative Deep Learning Models of Business Processes”
I suggest either to rewrite the latter as “Deep Learning models” as they are presented at line 42 or the former in a more specific manner
- Line 94: “i.e..” -> “i.e.,”
- Lines 124 to 128 are a bit convoluted. I would prefer to see it rephrased, in particular, the concept of hyperparameter optimisation is not new to the audience of this venue, therefore you could rephrase it by simply stating that: “in the post processing, SIMOD uses an hyperparameter optimisation technique to discover the SIMOD configuration maximising the ELS between the produced log and the ground truth”
- Line 191: Can you give a reference for the hallucination methodology?
- Figure 3: I would prefer to see a little incipit to the caption enumeration, instead of starting with the enumeration, I would prefer to see something like: “Explored network architectures: a) [..]”
- Table 1: why is the format of the last two columns different? They are both reporting duration, either use the hour/minute specification or the floating hour
- Table 1: Why are you using both average and mean in the columns “avg. activities per trace” and “mean duration”. Aren’t Average and Mean the same thing in statistics?
- Lines 222 and 223: Are these metrics a contribution of the work? if not, could you give a reference for them?
- Section 4.2 could you make bold the metrics in the respective paragraphs where you explain them? Like you did in section 2 and 3 with the various processing phases
- Line 260: could you give reference to the Hungarian algorithm?
- Line 288: what do you mean by predictable behaviour? Can you support this claim by some sort of metrics?
- Line 344 “Francescomarino” -> “Di Francescomarino”
- Line 351 “Rosa” -> “La Rosa”
- Line 364 “Francescomarino” -> “Di Francescomarino”
- Lines 370 and 374 incomplete reference missing publisher or at least venue
- Line 393 “Rosa” -> “La Rosa”

Reviewer 3 ·

Basic reporting

The paper is well organized; in fact, despite its structure deviates from the standard PeerJ template, I believe that this allows it to gain in readability.
The language used in it is professional, clear and unambiguous.

The figures and tables are readable and clear enough (see my detailed comments for some advice on how to improve Figure 1).

The paper is self contained, and it provides sufficient background information for understanding the proposal.

Relevant related work in the literature is covered, which allows for positioning the proposal adequately.

The auxiliary material made available for reproducibility purposes (raw data, code and results) can be accessed easily and opened correctly, complies with the journal's Data Sharing policy, and it looks well documented and explained.

Experimental design

The paper does not offer novel contributions from a technical/methodological viewpoint (e.g., in terms of novel algorithms, methodologies, quality metrics).

However, I do believe that it fits well the Aims and Scope of the journal, for it provides an nice empirical analysis of two alterative approaches to process trace generation/simulation (based on the construction of model-based process simulation models and of DL-based generative models, respectively).
Such an important comparative study is indeed lacking in the field.
The paper identifies this gap in the literature and clearly states the goal of filling it through a well defined experimentation. In fact, some of the experimental findings presented in the paper look pretty novel and interesting.

The data, methods and evaluation procedures adopted in the study are described in a complete enough way, but the addition of certain technical details could help improve the paper (see Section "General comments" for detailed comments on this respect).

Unfortunately, however, the scope of the study is somewhat limited, in terms of both the techniques analyzed and the datasets employed. See section "General comments" for a detailed discussion of this limitation and of possible ways to reduce it.

Validity of the findings

The experiments are fully replicable.

The testing procedure is correct, but not complete enough in my opinion, due to the lack of "recall-oriented" evaluation criteria (see section "General comments" for details on this point).

The discussion of the test results is articulated and clear, but it needs to be improved. In particular, I am afraid that certain conclusions are not fully supported by the results obtained (see section "General comments" for details).

Additional comments

*** PAPER SUMMARY
The paper presents a comparative empirical study of two alternative existing approaches to the (data-driven) generation of business process traces:
(A) building up a Data-Driven Simulation (DDS) model, and
(B) training a Deep Learning (DL) model for next-event prediction.
Specifically, for constructing DDS models the Authors adopt the pipeline procedure and the SIMOD toolset published in (Camargo et al., 2020), while they induce DL models by training different RNN-based architectures proposed in (Camargo et al., 2019) --more precisely, some variants of these architectures devised to generate both the start time and end time of the next activity, but not the role of their associated executors as in (Camargo et al., 2019).
The results of experiments performed on five public logs are presented, while using a number of ad hoc measures to evaluate the similarity between the traces generated and real ones.
The relative strengths of the two approaches are discussed, and the opportunity of developing hybrid approaches, capable of combine these strengths, is prospected.


*** EVALUATION
I appreciate the idea of conducting a comparative empirical analysis of model-based process simulation techniques and DL-based generative models, relatively to their capability to generate good-quality process traces.
Such an important kind of empirical study is indeed lacking in the field.
The paper is well organized and is easy to read, and its figures and tables are readable and clear enough.
The paper contains sufficient background information and references to relevant prior literature.
Auxiliary material (raw data, code and results) is of good quality and it can be opened correctly.

However, I am afraid that the quality of the work is undermined by a number of flaws, discussed below, which would need to be addressed appropriately.


1) Limited range of DL-based generative models analysed.
The induction of DL-based generative models for predicting/simulating process behaviors is a topical problem, for which there were recently proposed solutions, like those in (Lin et al., 2019) and (Taymouri et al., 2020).
In fact, you have mentioned these works, but excluded them all from the experimentation because of their "inability to predict timestamps and durations".

However, you might well decide to partially evaluate these methods by only using the CFLS metrics, provided that a public implementation is available for both methods --actually this seems to hold for the method in (Taymouri et al., 2020).

More importantly, it seems to me that the GAN-based method presented in (Taymouri et al., 2020) can predict/generate event timestamps, which can be used to compute activity durations (despite they are not returned directly by the method itself).
Indeed, you could train the model of (Taymouri et al., 2020) against traces containing a pair of instantaneous events for each of their activity instances (namely, a start event and a completion event for each activity instance), and then use the trained generator to produce traces having the same structure.
I presume that the computation of evaluation metrics concerning activity durations and cycle times can be easily adapted to traces of this form.

In my opinion, extending the experimentation to this method would add value to your empirical study, beside making it more complete.


2) Biased evaluation criteria.
The evaluation metrics computed in the tests only focus on precision-like aspects, so that one cannot assess whether the traces generated by a method really cover the variety of process behaviors that occur in the test log. In other words, there may well be modes of the test data distribution that can be missed!
I hence suggest that additional, recall-like, metrics be defined, quantifying how well the test traces are represented by (the sample traces yielded) each of the discovered simulation/generative models.


3) Limited range of datasets.
Three of the logs are very small and, thus, hardly contain sufficient data to train a deep model appropriately.
More importantly, I am afraid that all the processes considered in the experimentation are rather simple and pretty regular/structured.

Thus, it may be risky to draw, from your current experimental results, general conclusions on the relative strength of the two classes of process simulation approaches under analysis.
In particular, I am not fully convinced that DSS models are good (and better than DL-based ones) at capturing control-flow behaviors accurately.
I think, indeed, that such a claim would need to be substantiated by a wider range of tests, encompassing logs of more flexible and/or more complex processes (e.g., featuring long-distance activity dependencies).
Indeed, it may well be that DL models will capture the variety of control-flow behaviors of such processes better than DSS models hinging on local activity dependencies and local time distributions --as partly the CFLS scores that these methods obtained on the last log seem to suggest.

I do understand that it is difficult to find other public logs containing both start and end times for each event, but adding logs of less structured processes would really strengthen the value, coverage and significance of your empirical study.
To this respect, please consider the possibility to relax the requirement of dealing with traces containing only non instantaneous events (i.e., traces where each step encodes the execution of one activity lasting over the time, and associated with both start and end timestamps).
In principle, one could artificially keep all the activity durations fixed to 0, when trying to simulate a log that only stores the start or completion timestamp for each activity instance. It seems to me that the MAE and EMD scores would still make sense in such a simulation setting, and would give useful information on how accurately the model generates the event timestamps (and, indirectly, the cycle time).


4) Some conclusions look too strong (and not fully supported by test evidence).

4.a) In the concluding section you say that "DDS models are suitable for capturing the sequence of activities (or possibly other categorical attributes) of a process."
However, this claim does not seem to be not fully aligned to the test results.

Indeed, the CFLS score of GRU and LSTM models looks better than that of DDS on the real-life log BPI2017W, which is the only log containing more than 10K traces (and 100K events).
May it be the case that DL models capture activity sequence as accurately as (or even better than) DSS models when applied to complex enough processes (and provided sufficiently large amounts of training data)?

On the other hand, on the log of simple processes featuring few activities, DL models could be penalized by using embeddings for the activities (and low-dimensional categorical event attributes). There is, indeed, some evidence in the literature (Everman et al., 2019) that one-hot encoding schemes may help obtain more accurate models than embeddings in such a case --in principle, low-dimensional embeddings may lead to loss of useful information.
I suggest that you consider the possibility to test DL models obtained with one-hot encodings for the activities (and, possibly, for the users/roles), when the number of them is small.


4.b) Still in the concluding section, you say that "DL models outperform DDS models when predicting the timing of activities, specifically the waiting times between activities."
However, the assumption of eager-resources, related to the generation of waiting times, is rather rough.
In my opinion, some way of overcoming this limitation of your DSS models should be prospected, which does not necessarily rely on exploiting DL models.


5) Some lack of technical details and of critical discussion.
- Which specific families of PDFs are used to model inter-arrival times and activity durations?
- The discussion on hybrid approaches combining DSS and DL methods, in order to exploit the strengths of both synergistically, is rather shallow from a technical viewpoint. Some more technical reflections on this respects could add value to your work.


*** MINOR DETAILED NOTES.

In my opinion, Figure 1 is a little misleading.
The "post-processing" stage (lines 124-128) of the SIMOD pipeline looks more like a part of an optimization loop that also encompasses the "simulation" stage.
In other words, after initializing the DDS model in the "preprocessing" stage, one can iteratively refine it by generating a number of traces and optimizing the current parameters of the DDS, based on the quality of the traces generated.
As a matter of fact, Bayesian optimization does entail itself a number of simulation runs (actually, 5 runs for each of the 50 parameter configurations explored).
Please, try to describe more clearly this part of the SIMOD pipeline.

Lines 136-138:
It looks like you are presenting GANs and RNNs as two orthogonal solutions.
In fact, a GAN can include RNN components --this is indeed the case of the DL architecture proposed in (Taymouri et al., 2020)

Line 175:
The expression "duration of an event" may sound odd to readers who are used to consider process events as instantaneous events, related to lifecycle transitions of the activities --while the latter can last over time when executed.
Please, clarify this point and define explicitly the meaning of term "event" in your setting.

Figure 3:
Please, clarify the meaning of label "role prefixes".
It seems to me that roles are not used at all in your experimental framework --indeed, the term "role" never appears in Section 4. Anyway, no information is provided about how executors roles were defined/extracted for each log.
Please, clarify this point.
All of the three DL architectures shown in the figure should produce three kinds of output for each generated event: its type (activity), its start time and its end time.
Please make this explicit in the figure, in order to prevent the reader from interpreting that the three output dimensions correspond to the input ones (i.e., activity, role and times).

Line 229:
What is the cost of substitution operations?
Usually, in the computation of trace-to-model alignments, substitutions are disallowed, since substitutions may lead to too optimistic alignment measurements.

Line 230:
You say that the transposition of mutually concurrent activities is allowed for free (i.e. exchanging the occurrences of mutually concurrent activities costs 0).
How do you decide whether two activities are in parallel? Do you exploit some kind of oracle/expert or log-driven heuristics?

In principle, in the case of a DDS model, this information can be extracted from the control-flow structure of the DSS model itself. Do you use the same information for DL-based generative models (which do not make activity dependencies explicit)?
Or do you resort to classical log-driven ordering relationships, induced heuristics from the input log?

To be frank, I believe that using log-driven relationships is definitely reasonable when building/optimizing a DSS model.
However, I am not convinced that it makes sense when it comes to evaluating the quality of simulation results.
In such a case, indeed, log-driven relationships should be validated preliminary by an expert (possibly yourselves) or, at least, by using some well-defined criterion for automatically assessing the significance of these relationships (and filtering out those that are not significant enough) or for making the cost of exchanging two activities X and Y depend on how likely X and Y seem to be in parallel.
It could be beneficial for the reader to have this point clarified (and discussed appropriately) in the revised version of the manuscript.


Line 258:
Please add details on the penalty mechanism.

Line 265:
"holdup" -> "hold-out"

---

## Round 0.2 · accepted · Accept

The article presents an extensive revision of a previously submitted paper, which addresses all the observations made by the reviewers in a fully satisfactory manner.

Please take into account the few minor comments and typos identified by the reviewers when preparing the proof.

Reviewer 1 ·

Basic reporting

Basic reporting
The article presents an empirical comparison of two approaches for process discovery/predictive monitoring. The first approach is based on simulation mining: a detailed simulation process model (e.g., a Petri net) is discovered from the data. Then, the model is enriched and used for the analysis. The second approach is based on the deep learning - a generative neural network is trained from the data. Both models are essentially trace generators (including rich features that can be generated into the future). The empirical evaluation is very nice and the topic is worth investigating. The paper is very relevant for practitioners and scientists who are working in the area of process mining. One of the novelties of the approach is the realizations that the two representations (simulation and deep learning) have equivalent usages.

In terms of the standards: the paper is well-written, the literature is covered in a sufficient manner, the paper is structured professionally, is self-contained, and the results are presented in a satisfying manner.

Experimental design

The experimental results are well-aligned with the journal's Aims and Scope. The research questions are well defined, and the paper does fill in a gap in the literature. It is the first paper that considers this equivalence between two generative families of models. I am happy with the update to the paper.

Validity of the findings

I am satisfied with the validity of the findings, after reading the comments by the authors.

Conclusions and limitations are well-stated and the empirical evaluation is described in a valid and satisfying manner.

Additional comments

Excellent rework of the paper.

Reviewer 2 ·

Basic reporting

I have a couple of minor comments which have to be considered only as a refinement of the already well done work.

Line 196: approach or approaches?

Figure 1: "interarrival dist." and "activities dist." could be written extensively (if you have enough space)
Figure 1: "Resource avail calendar" -> "Resource avail. calendar"

Figure 3 top left both in diagram (a) and (b): "role category" -> "roles category"

Figure 4.b bottom right: "sufixes"-> "suffixes"

Experimental design

no comment

Validity of the findings

no comment

Additional comments

The authors properly addressed all my concerns regarding their work.

Reviewer 3 ·

Basic reporting

No comment. See section "General comments for the Author".

Experimental design

No comment. See section "General comments for the Author".

Validity of the findings

No comment. See section "General comments for the Author".

Additional comments

The quality of the work is good, in terms of readability, novelty, relevance, technical depth, experimental design and discussion of the experimental results --as I already noticed in my first-round review.

I believe that the Authors have overcome the flaws that affected the former version of the manuscript.
In particular, I am fully satisfied with how they have addressed my comments.

Please notice that there is something wrong with text formatting in page 12:
It looks like the title and beginning of Section 7 have been written within footnote 6.